# Stand-Alone Self-Attention in Vision Models

**Prajit Ramachandran**[*]        **Niki Parmar**[*]        **Ashish Vaswani**[*]

**Irwan Bello**        **Anselm Levskaya**[†]        **Jonathon Shlens**

Google Research, Brain Team
`{prajit, nikip, avaswani}@google.com`

## Abstract

Convolutions are a fundamental building block of modern computer vision systems. Recent approaches have argued for going beyond convolutions in order to capture long-range dependencies. These efforts focus on augmenting convolutional models with *content-based* interactions, such as self-attention and non-local means, to achieve gains on a number of vision tasks. The natural question that arises is whether attention can be a stand-alone primitive for vision models instead of serving as just an augmentation on top of convolutions. In developing and testing a pure self-attention vision model, we verify that self-attention can indeed be an effective stand-alone layer. A simple procedure of replacing all instances of spatial convolutions with a form of self-attention applied to ResNet model produces a fully self-attentional model that outperforms the baseline on ImageNet classification with 12% fewer FLOPS and 29% fewer parameters. On COCO object detection, a pure self-attention model matches the mAP of a baseline RetinaNet while having 39% fewer FLOPS and 34% fewer parameters. Detailed ablation studies demonstrate that self-attention is especially impactful when used in later layers. These results establish that stand-alone self-attention is an important addition to the vision practitioner's toolbox. Code for this project is made available.[1]

## 1   Introduction

Digital image processing arose from the recognition that handcrafted linear filters applied convolutionally to pixelated imagery may subserve a large variety of applications [1]. The success of digital image processing as well as biological considerations [2, 3] inspired early practitioners of neural networks to exploit convolutional representations in order to provide parameter-efficient architectures for learning representations on images [4, 5].

The advent of large datasets [6] and compute resources [7] made convolution neural networks (CNNs) the backbone for many computer vision applications [8–10]. The field of deep learning has in turn largely shifted toward the design of architectures of CNNs for improving the performance on image recognition [11–16], object detection [17–19] and image segmentation [20–22]. The translation equivariance property of convolutions has provided a strong motivation for adopting them as a building block for operating on images [23, 24]. However, capturing long range interactions for convolutions is challenging because of their poor scaling properties with respect to large receptive fields.

---

[*]Denotes equal contribution. Ordering determined by random shuffle.
[†]Work done as a member of the Google AI Residency Program.
[1] https://github.com/google-research/google-research/tree/master/standalone_self_attention_in_vision_models

The problem of long range interactions has been tackled in sequence modeling through the use of attention. Attention has enjoyed rich success in tasks such as language modeling [25, 26], speech recognition [27, 28] and neural captioning [29]. Recently, attention modules have been employed in discriminative computer vision models to boost the performance of traditional CNNs. Most notably, a channel-based attention mechanism termed Squeeze-Excite may be applied to selectively modulate the scale of CNN channels [30, 31]. Likewise, spatially-aware attention mechanisms have been used to augment CNN architectures to provide contextual information for improving object detection [32] and image classification [33–35]. These works have used global attention layers as an add-on to existing convolutional models. This global form attends to all spatial locations of an input, limiting its usage to small inputs which typically require significant downsampling of the original image.

In this work, we ask the question if content-based interactions can serve as the primary primitive of vision models instead of acting as an augmentation to convolution. To this end, we develop a simple local self-attention layer that can be used for both small and large inputs. We leverage this stand-alone attention layer to build a fully attentional vision model that outperforms the convolutional baseline for both image classification and object detection while being parameter and compute efficient. Furthermore, we conduct a number of ablations to better understand stand-alone attention. We hope that this result will spur new research directions focused on exploring content-based interactions as a mechanism for improving vision models.

## 2 Background

### 2.1 Convolutions

Convolutional neural networks (CNNs) are typically employed with small neighborhoods (i.e. kernel sizes) to encourage the network to learn local correlation structures within a particular layer. Given an input $x \in \mathbb{R}^{h \times w \times d_{in}}$ with height $h$, width $w$, and input channels $d_{in}$, a local neighborhood $\mathcal{N}_k$ around a pixel $x_{ij}$ is extracted with spatial extent $k$, resulting in a region with shape $k \times k \times d_{in}$ (see Figure 1).

Given a learned weight matrix $W \in \mathbb{R}^{k \times k \times d_{out} \times d_{in}}$, the output $y_{ij} \in \mathbb{R}^{d_{out}}$ for position $ij$ is defined by spatially summing the product of depthwise matrix multiplications of the input values:

$$y_{ij} = \sum_{a,b \in \mathcal{N}_k(i,j)} W_{i-a,j-b}\, x_{ab} \tag{1}$$

where $\mathcal{N}_k(i,j) = \left\{ a,b \mid |a-i| \leq k/2, |b-j| \leq k/2 \right\}$ (see Figure 2). Importantly, CNNs employ *weight sharing*, where $W$ is reused for generating the output for all pixel positions $ij$. Weight sharing enforces translation equivariance in the learned representation and consequently decouples the parameter count of the convolution from the input size.

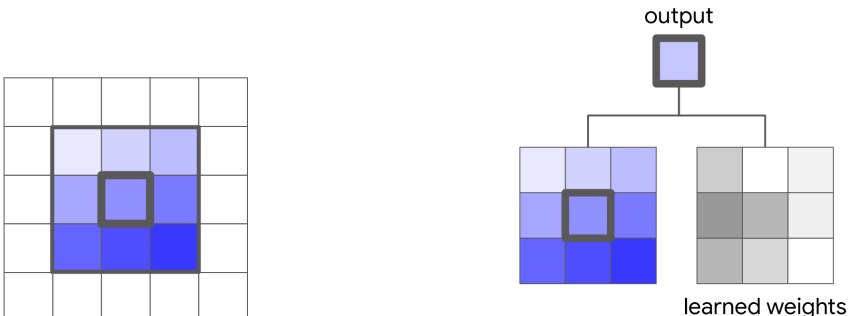

Figure 1: An example of a local window around $i = 3, j = 3$ (one-indexed) with spatial extent $k = 3$.

Figure 2: An example of a $3 \times 3$ convolution. The output is the inner product between the local window and the learned weights.

A wide array of machine learning applications have leveraged convolutions to achieve competitive results including text-to-speech [36] and generative sequence models [37, 38]. Several efforts have

reformulated convolutions to improve the predictive performance or the computational efficiency of a model. Notably, depthwise-separable convolutions provide a low-rank factorization of spatial and channel interactions [39–41]. Such factorizations have allowed for the deployment of modern CNNs on mobile and edge computing devices [42, 43]. Likewise, relaxing translation equivariance has been explored in locally connected networks for various vision applications [44].

## 2.2 Self-Attention

Attention was introduced by [45] for the encoder-decoder in a neural sequence transduction model to allow for content-based summarization of information from a variable length source sentence. The ability of attention to learn to focus on important regions within a context has made it a critical component in neural transduction models for several modalities [26, 29, 27]. Using attention as a primary mechanism for representation learning has seen widespread adoption in deep learning after [25], which entirely replaced recurrence with self-attention. Self-attention is defined as attention applied to a single context instead of across multiple contexts (in other words, the query, keys, and values, as defined later in this section, are all extracted from the same context). The ability of self-attention to directly model long-distance interactions and its parallelizability, which leverages the strengths of modern hardware, has led to state-of-the-art models for various tasks [46–51].

An emerging theme of augmenting convolution models with self-attention has yielded gains in several vision tasks. [32] show that self-attention is an instantiation of non-local means [52] and use it to achieve gains in video classification and object detection. [53] also show improvements on image classification and achieve state-of-the-art results on video action recognition tasks with a variant of non-local means. Concurrently, [33] also see significant gains in object detection and image classification through augmenting convolutional features with global self-attention features. This paper goes beyond [33] by removing convolutions and employing local self-attention across the entirety of the network. Another concurrent work [35] explores a similar line of thinking by proposing a new content-based layer to be used across the model. This approach is complementary to our focus on directly leveraging existing forms of self-attention for use across the vision model.

We now describe a stand-alone self-attention layer that can be used to replace spatial convolutions and build a fully attentional model. The attention layer is developed with a focus on simplicity by reusing innovations explored in prior works, and we leave it up to future work to develop novel attentional forms.

Similar to a convolution, given a pixel $x_{ij} \in \mathbb{R}^{d_{in}}$, we first extract a local region of pixels in positions $ab \in \mathcal{N}_k(i, j)$ with spatial extent $k$ centered around $x_{ij}$, which we call the *memory block*. This form of local attention differs from prior work exploring attention in vision which have performed global (i.e., all-to-all) attention between all pixels [32, 33]. Global attention can only be used after significant spatial downsampling has been applied to the input because it is computationally expensive, which prevents its usage across all layers in a fully attentional model.

Single-headed attention for computing the pixel output $y_{ij} \in \mathbb{R}^{d_{out}}$ is then computed as follows (see Figure 3):

$$y_{ij} = \sum_{a,b \in \mathcal{N}_k(i,j)} \texttt{softmax}_{ab} \left( q_{ij}^\top k_{ab} \right) v_{ab} \tag{2}$$

where the *queries* $q_{ij} = W_Q x_{ij}$, *keys* $k_{ab} = W_K x_{ab}$, and *values* $v_{ab} = W_V x_{ab}$ are linear transformations of the pixel in position $ij$ and the neighborhood pixels. $\texttt{softmax}_{ab}$ denotes a softmax applied to all logits computed in the neighborhood of $ij$. $W_Q, W_K, W_V \in \mathbb{R}^{d_{out} \times d_{in}}$ are all learned transforms. While local self-attention aggregates spatial information over neighborhoods similar to convolutions (Equation 1), the aggregation is done with a convex combination of value vectors with mixing weights ($\texttt{softmax}_{ab}(\cdot)$) parametrized by content interactions. This computation is repeated for every pixel $ij$. In practice, multiple attention *heads* are used to learn multiple distinct representations of the input. It works by partitioning the pixel features $x_{ij}$ depthwise into $N$ groups $x_{ij}^n \in \mathbb{R}^{d_{in}/N}$, computing single-headed attention on each group separately as above with different transforms $W_Q^n, W_K^n, W_V^n \in \mathbb{R}^{d_{in} \times d_{out}/N}$ per head, and then concatenating the output representations into the final output $y_{ij} \in \mathbb{R}^{d_{out}}$.

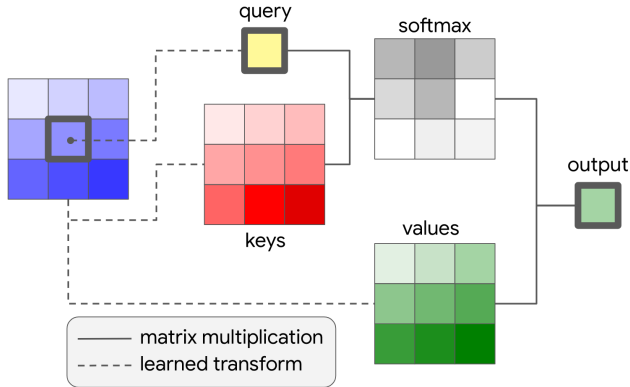

Figure 3: An example of a local attention layer over spatial extent of $k = 3$.

Figure 4: An example of relative distance computation. The relative distances are computed with respect to the position of the highlighted pixel. The format of distances is *row offset*, *column offset*.

As currently framed, no positional information is encoded in attention, which makes it permutation equivariant, limiting expressivity for vision tasks. Sinusoidal embeddings based on the absolute position of pixels in an image $(ij)$ can be used [25], but early experimentation suggested that using relative positional embeddings [51, 46] results in significantly better accuracies. Instead, attention with 2D relative position embeddings, *relative attention*, is used. Relative attention starts by defining the relative distance of $ij$ to each position $ab \in \mathcal{N}_k(i, j)$. The relative distance is factorized across dimensions, so each element $ab \in \mathcal{N}_k(i, j)$ receives two distances: a row offset $a - i$ and column offset $b - j$ (see Figure 4). The row and column offsets are associated with an embedding $r_{a-i}$ and $r_{b-j}$ respectively each with dimension $\frac{1}{2}d_{out}$. The row and column offset embeddings are concatenated to form $r_{a-i,b-j}$. This spatial-relative attention is now defined as

$$y_{ij} = \sum_{a,b \in \mathcal{N}_k(i,j)} \texttt{softmax}_{ab} \left( q_{ij}^\top k_{ab} + q_{ij}^\top r_{a-i,b-j} \right) v_{ab} \tag{3}$$

Thus, the logit measuring the similarity between the query and an element in $\mathcal{N}_k(i, j)$ is modulated both by the content of the element and the relative distance of the element from the query. Note that by infusing relative position information, self-attention also enjoys translation equivariance, similar to convolutions.

The parameter count of attention is independent of the size of spatial extent, whereas the parameter count for convolution grows quadratically with spatial extent. The computational cost of attention also grows slower with spatial extent compared to convolution with typical values of $d_{in}$ and $d_{out}$. For example, if $d_{in} = d_{out} = 128$, a convolution layer with $k = 3$ has the same computational cost as an attention layer with $k = 19$.

## 3  Fully Attentional Vision Models

Given a local attention layer as a primitive, the question is how to construct a fully attentional architecture. We achieve this in two steps:

### 3.1  Replacing Spatial Convolutions

A spatial convolution is defined as a convolution with spatial extent $k > 1$. This definition excludes $1 \times 1$ convolutions, which may be viewed as a standard fully connected layer applied to each pixel independently.[2] This work explores the straightforward strategy of creating a fully attentional vision model: take an existing convolutional architecture and replace every instance of a spatial convolution with an attention layer. A $2 \times 2$ average pooling with stride 2 operation follows the attention layer whenever spatial downsampling is required.

This work applies the transform on the ResNet family of architectures [15]. The core building block of a ResNet is a *bottleneck block* with a structure of a $1 \times 1$ down-projection convolution, a $3 \times 3$ spatial convolution, and a $1 \times 1$ up-projection convolution, followed by a residual connection between the input of the block and the output of the last convolution in the block. The bottleneck block is repeated multiple times to form the ResNet, with the output of one bottleneck block being the input of the next bottleneck block. The proposed transform swaps the $3 \times 3$ spatial convolution with a self-attention layer as defined in Equation 3. All other structure, including the number of layers and when spatial downsampling is applied, is preserved. This transformation strategy is simple but possibly suboptimal. Crafting the architecture with attention as a core component, such as with architecture search [54], holds the promise of deriving better architectures.

### 3.2   Replacing the Convolutional Stem

The initial layers of a CNN, sometimes referred to as the *stem*, play a critical role in learning local features such as edges, which later layers use to identify global objects. Due to input images being large, the stem typically differs from the core block, focusing on lightweight operations with spatial downsampling [11, 15]. For example, in a ResNet, the stem is a $7 \times 7$ convolution with stride 2 followed by $3 \times 3$ max pooling with stride 2.

At the stem layer, the content is comprised of RGB pixels that are individually uninformative and heavily spatially correlated. This property makes learning useful features such as edge detectors difficult for content-based mechanisms such as self-attention. Our early experiments verify that using self-attention form described in Equation 3 in the stem underperforms compared to using the convolution stem of ResNet.

The distance based weight parametrization of convolutions allows them to easily learn edge detectors and other local features necessary for higher layers. To bridge the gap between convolutions and self-attention while not significantly increasing computation, we inject distance based information in the pointwise $1 \times 1$ convolution ($W_V$) through spatially-varying linear transformations. The new value transformation is $\tilde{v}_{ab} = \left( \sum_m p(a,b,m) W_V^m \right) x_{ab}$ where multiple value matrices $W_V^m$ are combined through a convex combination of factors that are a function of the position of the pixel in its neighborhood $p(a,b,m)$. The position dependent factors are similar to convolutions, which learn scalar weights dependent on the pixel location in a neighborhood. The stem is then comprised of the attention layer with spatially aware value features followed by max pooling. For simplicity, the attention receptive field aligns with the max pooling window. More details on the exact formulation of $p(a,b,m)$ is given in the appendix.

## 4   Experiments

### 4.1   ImageNet Classification

**Setup**   We perform experiments on ImageNet classification task [55] which contains 1.28 million training images and 50000 test images. The procedure described in Section 3.1 of replacing the spatial convolution layer with a self-attention layer from inside each bottleneck block of a ResNet-50 [15] model is used to create the attention model. The multi-head self-attention layer uses a spatial extent of $k = 7$ and 8 attention heads. The position-aware attention stem as described above is used. The stem performs self-attention within each $4 \times 4$ spatial block of the original image, followed by batch normalization and a $4 \times 4$ max pool operation. Exact hyperparameters can be found in the appendix.

To study the behavior of these models with different computational budgets, we scale the model either by width or depth. For width scaling, the base width is linearly multiplied by a given factor across all layers. For depth scaling, a given number of layers are removed from each *layer group*. There are 4 layer groups, each with multiple layers operating on the same spatial dimensions. Groups are delineated by spatial downsampling. The 38 and 26 layer models remove 1 and 2 layers respectively from each layer group compared to the 50 layer model.

**Results**   Table 1 and Figure 5 shows the results of the full attention variant compared with the convolution baseline. Compared to the ResNet-50 baseline, the full attention variant achieves $0.5\%$

|  | ResNet-26 | | | ResNet-38 | | | ResNet-50 | | |
|---|---|---|---|---|---|---|---|---|---|
|  | FLOPS (B) | Params (M) | Acc. (%) | FLOPS (B) | Params (M) | Acc. (%) | FLOPS (B) | Params (M) | Acc. (%) |
| **Baseline** | 4.7 | 13.7 | 74.5 | 6.5 | 19.6 | 76.2 | 8.2 | 25.6 | 76.9 |
| **Conv-stem + Attention** | 4.5 | 10.3 | **75.8** | 5.7 | 14.1 | **77.1** | 7.0 | 18.0 | **77.4** |
| **Full Attention** | 4.7 | 10.3 | 74.8 | 6.0 | 14.1 | 76.9 | 7.2 | 18.0 | **77.6** |

Table 1: ImageNet classification results for a ResNet network with different depths. *Baseline* is a standard ResNet, *Conv-stem + Attention* uses spatial convolution in the stem and attention everywhere else, and *Full Attention* uses attention everywhere including the stem. The attention models outperform the baseline across all depths while having 12% fewer FLOPS and 29% fewer parameters.

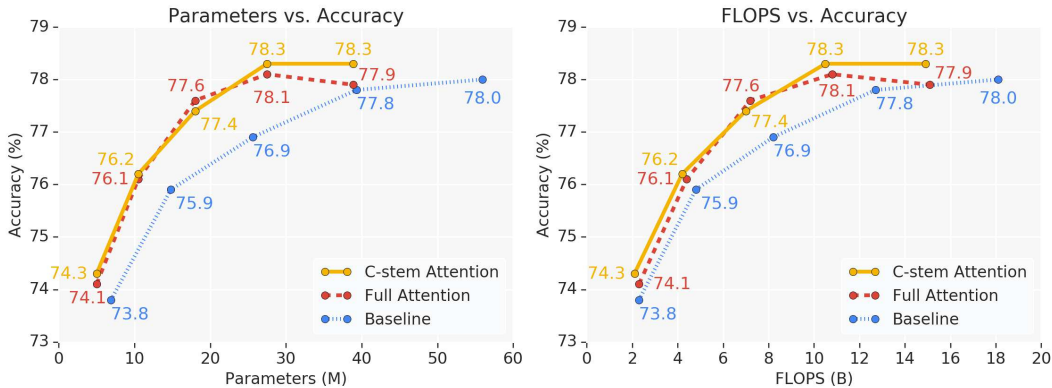

Figure 5: Comparing parameters and FLOPS against accuracy on ImageNet classification across a range of network widths for ResNet-50. Attention models have fewer parameters and FLOPS while improving upon the accuracy of the baseline.

higher classification accuracy while having 12% fewer floating point operations (*FLOPS*)[3] and 29% fewer parameters. Furthermore, this performance gain is consistent across most model variations generated by both depth and width scaling.

## 4.2   COCO Object Detection

**Setup**    In this section, we evaluate attention models on the COCO object detection task [56] using the RetinaNet architecture [18]. RetinaNet is an object detection model that consists of a *backbone* image classification network followed by a Feature Pyramid Network (*FPN*) [57] and two output networks known as detection heads. We experiment with making the backbone and/or the FPN and detection heads fully attentional. The backbone models are the same models described in Section 4.1. The details of how the FPN and detection heads are made fully attentional are provided in the appendix.

**Results**    Table 2 shows the object detection results. Using an attention-based backbone in the RetinaNet matches the mAP of using the convolutional backbone but contains 22% fewer parameters. Furthermore, employing attention across all parts of the model including the backbone, FPN, and detection heads matches the mAP of the baseline RetinaNet while using 34% fewer parameters and 39% fewer FLOPS. These results demonstrate the efficacy of stand-alone attention across multiple vision tasks.

| Detection Heads + FPN | Backbone | FLOPS (B) | Params (M) | mAP$_{coco/50/75}$ | mAP$_{s/m/l}$ |
|---|---|---|---|---|---|
| Convolution | Baseline | 182 | 33.4 | 36.5 / 54.3 / 39.0 | 18.3 / 40.6 / 51.7 |
| | Conv-stem + Attention | 173 | 25.9 | 36.8 / 54.6 / 39.3 | 18.4 / 41.1 / 51.7 |
| | Full Attention | 173 | 25.9 | 36.2 / 54.0 / 38.7 | 17.5 / 40.3 / 51.7 |
| Attention | Conv-stem + Attention | **111** | **22.0** | 36.6 / 54.3 / 39.1 | 19.0 / 40.7 / 51.1 |
| | Full Attention | **110** | **22.0** | 36.6 / 54.5 / 39.2 | 18.5 / 40.6 / 51.6 |

Table 2: Object detection on COCO dataset with RetinaNet [18]. Mean Average Precision (mAP) is reported at three different IoU values and for three different object sizes (small, medium, large). The fully attentional models achieve similar mAP as the baseline while having up to 39% fewer FLOPS and 34% fewer parameters.

| Conv Groups | Attention Groups | FLOPS (B) | Params (M) | Top-1 Acc. (%) |
|---|---|---|---|---|
| - | 1, 2, 3, 4 | 7.0 | 18.0 | 80.2 |
| 1 | 2, 3, 4 | 7.3 | 18.1 | **80.7** |
| 1, 2 | 3, 4 | 7.5 | 18.5 | **80.7** |
| 1, 2, 3 | 4 | 8.0 | 20.8 | 80.2 |
| 1, 2, 3, 4 | - | 8.2 | 25.6 | 79.5 |
| 2, 3, 4 | 1 | 7.9 | 25.5 | 79.7 |
| 3, 4 | 1, 2 | 7.8 | 25.0 | 79.6 |
| 4 | 1, 2, 3 | 7.2 | 22.7 | 79.9 |

Table 3: Modifying which layer groups use which primitive. Accuracies computed on validation set. The best performing models use convolutions for early groups and attention for later groups.

| Spatial Extent ($k \times k$) | FLOPS (B) | Top-1 Acc. (%) |
|---|---|---|
| $3 \times 3$ | 6.6 | 76.4 |
| $5 \times 5$ | 6.7 | 77.2 |
| $7 \times 7$ | 7.0 | 77.4 |
| $9 \times 9$ | 7.3 | 77.7 |
| $11 \times 11$ | 7.7 | 77.6 |

Table 4: Varying the spatial extent $k$. Parameter count is constant across all variations. Small $k$ perform poorly, but the improvements of larger $k$ plateaus off.

## 4.3 Where is stand-alone attention most useful?

The impressive performance of fully attentional models verifies that stand-alone attention is a viable primitive for vision models. In this section, we study which parts of the network benefit the most from stand-alone attention.

**Stem** First, we compare the performance of the attention stem against the convolution stem used in ResNet. All other spatial convolutions are replaced with stand-alone attention. Tables 1 and 2 and Figure 5 show the results on ImageNet classification and COCO object detection. For classification, the convolution stem consistently matches or outperforms the attention stem. For object detection, the convolution stem performs better when a the detection heads and FPN are also convolutional, but performs similarly when the entire rest of the network is fully attentional. These results suggest that convolutions consistently perform well when used in the stem.

**Full network** Next, we experiment with using convolution and stand-alone attention in different layer groups in a ResNet with a convolution stem. Table 3 shows that the best performing models use convolutions in the early groups and attention in the later groups. These models are also similar in terms of FLOPS and parameters to the fully attentional model. In contrast, when attention is used in the early groups and convolutions are used in the later groups, the performance degrades despite a large increase in the parameter count. This suggests that convolutions may better capture low level features while stand-alone attention layers may better integrate global information.

Taken together, these results suggest that vision practitioners should focus on developing strategies of designing architectures that combine the comparative advantages of convolution and stand-alone attention.

| Positional Encoding Type | FLOPS (B) | Params (M) | Top-1 Acc. (%) |
|---|---|---|---|
| none | 6.9 | 18.0 | 77.6 |
| absolute | 6.9 | 18.0 | 78.2 |
| relative | 7.0 | 18.0 | 80.2 |

Table 5: The effect of changing the positional encoding type for attention. Accuracies computed on the validation set. Relative encodings significantly outperform other strategies.

| Attention Type | FLOPS (B) | Params (M) | Top-1 Acc. (%) |
|---|---|---|---|
| $q^\top r$ | 6.1 | 16.7 | 76.9 |
| $q^\top k + q^\top r$ | 7.0 | 18.0 | 77.4 |

Table 6: The effect of removing the $q^\top k$ interactions in attention. Using just $q^\top r$ interactions only drops accuracy by $0.5\%$.

| Attention Stem Type | FLOPS (B) | Top-1 Acc. (%) |
|---|---|---|
| stand-alone | 7.1 | 76.2 |
| spatial convolution for values | 7.4 | 77.2 |
| spatially aware values | 7.2 | **77.6** |

Table 7: Ablating the form of the attention stem. Spatially-aware value attention outperforms both stand-alone attention and values generated by a spatial convolution.

## 4.4 Which components are important in attention?

This section presents ablations designed to understand the contributions of the various components in the local attention layer. Unless specified, all attention models in the ablations use the convolution stem.

### 4.4.1 Effect of spatial extent of self-attention

The value of the spatial extent $k$ controls the size of the region each pixel can attend to. Table 4 studies the effect of varying the spatial extent. While using small $k$, such as $k = 3$, has a large negative impact on performance, the improvements of using a larger $k$ plateau around $k = 11$. The exact plateau value likely depends on specific settings of hyperparameters such as the feature size and number of attention heads used.

### 4.4.2 Importance of positional information

Table 5 ablates the different types of positional encodings that can be used: no positional encoding, a sinusodial encoding dependent on the absolute position of a pixel [25], and relative position encodings. Using any notion of positional encoding is beneficial over using none, but the type of positional encoding is also important. Relative position encodings perform $2\%$ better than absolute encodings. Furthermore, Table 6 demonstrates the important role of the content-relative interactions $(q \cdot r)$ in attention. Removing the content-content $(q \cdot k)$ interactions and just using the content-relative interactions drops the accuracy by only $0.5\%$. The importance of positional information suggests that future work may improve attention by exploring different parameterizations and usages of positional information.

### 4.4.3 Importance of spatially-aware attention stem

Table 7 compares using stand-alone attention in the stem with the attention stem with spatially-aware values proposed in Section 3.2. The proposed attention stem outperforms stand-alone attention by $1.4\%$ despite having a similar number of FLOPS, validating the utility of modifying attention for use in the stem. Furthermore, applying a spatial convolution to the values instead of a spatially-aware mixture of point-wise transformations proposed in Section 3.2 incurs more FLOPS and performs slightly worse. Future work can focus on unifying the spatially-aware attention used in the stem with the attention used in the main trunk of the network.

# 5  Discussion

In this work, we verified that content-based interactions can indeed serve as the primary primitive of vision models. A fully attentional network based off of the proposed stand-alone local self-attention layer achieves competitive predictive performance on ImageNet classification and COCO object detection tasks while requiring fewer parameters and floating point operations than the corresponding convolution baselines. Furthermore, ablations show that attention is especially effective in the later parts of the network.

We see several opportunities for improving the performance of these networks. First, the attention mechanism may be improved by developing better methods for capturing geometries [58, 59]. Second, the architectures employed for image classification and object detection were developed by applying a simple transformation to models designed for the convolutional primitive [13, 19]. It may be possible to achieve improvements by specifically searching for the architecture with an attention layer as a component in the design search space [31, 16, 21, 60]. Finally, additional work on proposing new attention forms that can capture low level features can make attention effective in the early layers of networks [61, 62].

Although the training efficiency and computational demand of an attention based architecture is favorable to a traditional convolution, the resulting network is slower in wall-clock time. The reason for this discrepancy is the lack of optimized kernels available on various hardware accelerators. In principle, depending on the degree to which the field deems that attention provides a viable path, it may be possible to significantly speed up the wall-clock time for training and inference accordingly.

While this work primarily focuses on content-based interactions to establish their virtue for vision tasks, in the future, we hope to unify convolution and self-attention to best combine their unique advantages. Given the success of content-based interactions on core computer vision tasks, we expect that future work may explore how attention could be applied to other vision tasks such as semantic segmentation [63], instance segmentation [64], keypoint detection [65], human pose estimation [66, 67] and other tasks currently addressed with convolutional neural networks.

### Acknowledgments

We thank Blake Hechtman, Justin Gilmer, Pieter-jan Kindermans, Quoc Le, Samy Bengio, and Shibo Wang for fruitful discussions and assistance with implementations as well as the larger Google Brain team for support and assistance.

## Footnotes

[2]Many deep learning libraries internally translate a $1 \times 1$ convolution to a simple matrix multiplication.

[3]Some prior works define a FLOP as a single atomic Multiply-Add, whereas we treat the Multiply and Add as 2 FLOPS. This causes a $2\times$ discrepancy in the reported number.

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
