[Supplementary Material · Stand_Alone_Self_Attention_in_Vision_Models_Appendix.pdf]

# A  Appendix

## A.1  Attention Stem

In this section, we first describe the standard self-attention layer followed by the spatially-aware mixtures in the attention stem.

For an input with $x_{ij} \in \mathbb{R}^{d_{in}}$ we define a standard single-headed self-attention layer as

$$q_{ij} = W_Q x_{ij} \tag{4}$$
$$k_{ij} = W_K x_{ij} \tag{5}$$
$$v_{ij} = W_V x_{ij} \tag{6}$$
$$y_{ij} = \sum_{a,b \in \mathcal{N}_k(i,j)} \mathtt{softmax}_{ab} \left( q_{ij}^\top k_{ab} \right) v_{ab} \tag{7}$$

where $W_Q, W_K, W_V \in \mathbb{R}^{d_{in} \times d_{out}}$ and the neighborhood $\mathcal{N}_k(i,j) = \{a,b \mid |a-i| \leq k/2, |b-j| \leq k/2\}$ yielding the intermediate per-pixel queries, keys, and values $q_{ij}, k_{ij}, v_{ij} \in \mathbb{R}^{d_{out}}$ and the final output $y_{ij} \in \mathbb{R}^{d_{out}}$.

The attention stem replaces the pointwise values $v_{ij}$ by spatially-aware linear transformations. For simplicity, we align the query, key and value receptive field with the max-pooling receptive field of $4 \times 4$. Then to inject distance aware value features, we use a convex combination of multiple value matrices $W_V^m$ where the combination weights are a function of the absolute position of the value in the pooling window. The functional form is defined in Equation 9 which computes the logit between the absolute embedding and the mixture embedding $\nu^m$.

$$v_{ab} = \sum_m p(a, b, m) W_V^m x_{ab} \tag{8}$$

$$p(a, b, m) = \mathtt{softmax}_m \left( (\mathrm{emb}_{row}(a) + \mathrm{emb}_{col}(b)) \cdot \nu^m \right) \tag{9}$$

Where $\mathrm{emb}_{row}(a)$ and $\mathrm{emb}_{col}(b)$ are pooling-window aligned row and column embeddings and $\nu^m$ is a per-mixture embedding. The resulting $p_{ab}^m$ are shared across the 4 attention heads for the mixture stem layer.

## A.2  ImageNet Training Details

For tuning, a validation set containing a $4\%$ random subset of the training set is used. Training is performed for 130 epochs using Nesterov's Accelerated Gradient [68, 69] with a learning rate of 1.6 which is linearly warmed up for 10 epochs followed by cosine decay [70]. A total batch size of 4096 is spread across 128 Cloud TPUv3 cores [71]. The setup uses batch normalization [40] with decay 0.9999 and exponential moving average with weight 0.9999 over trainable parameters [72, 73].

## A.3  Object Detection Training Details

The fully attentional object detection architecture uses the fully attentional classification models detailed in Section 4.1 as its backbone network. The rest of the architecture is obtained by replacing the $3 \times 3$ convolutions in the original RetinaNet architecture with self-attention layers of the same width ($d_{out} = 256$). We additionally apply $2 \times 2$ average pooling with stride 2 when replacing a strided convolution. The classification and regression heads share weights across all levels and their $W_V$ matrices are initialized randomly from a normal distribution with standard deviation 0.01 as in the original RetinaNet architecture [18]. Finally, we add an extra pointwise convolution at the end of the classification and box regression heads to mix the attentional heads. All self-attention layers use a spatial extent of $k = 7$ and 8 heads as for the image classification experiments.

We follow a similar training setup as in [18, 33]. All networks are trained for 150 epochs with a batch size of 64. The learning rate is warmed up linearly from 0 to 0.12 for one epoch and then decayed using a cosine schedule. We apply multiscale jitter, crop to a max dimension of 640 during training and randomly flip images horizontally with 50% probability.