[Reviews · NeurIPS 2019]

Reviewer 1



This paper proposes to build deep neural networks for computer vision based on fully self-attention layers. A local self-attention layer is proposed to overcome the limitations of the global attention layers that must be applied to reduced versions of the input image due to its computational load. This self-attention layer is used to replace all the convolutional layers in ResNet architectures. The proposed model is compared to standard CNN Resnet on ImageNet and Coco databases. Using self-attention layers allows to reduce the number of parameters of the model while maintaining the performance level. Ablation studies are conducted on different aspects of the proposed model. The goal of the replacing convolutions with local self-attention is a bit in contradiction with the initial objective of using attention layers. As noted by the authors, attention layers were introduced to take into account long term dependencies, what convolution layers could not do. An alternative direction is to use recurrent layers. Reducing self-attention to a local attention layer disable this long term dependency effect. Moreover, using multiple heads makes self-attention even closer to convolutions : it is local and learns multiple representation of the input. The paper could have studied more deeply the differences, both computationally and in terms of expected properties, between the local self-attention layers and convolutional layers. In the end, it seems that the parameters of the convolutional layer (the different kernel parameters) have been replaced by the parameters of the linear transformation Wq, Wk and Wv. The importance of positional features, shown in Table 5, could also have been more explored, because this is one of the main differences with CNN. Why is it important for self-attention ? Is it also useful for CNN on images ? (it has been used for CNN on language). Results in Table 6 are really surprising. One could consider that using only the positional interaction is a degenerated form of convolution, where each position in the kernel are encoded with a different set of parameters.

Reviewer 2



This paper answers the question on whether self-attention can be used as a stand-alone primitive for many vision tasks. The paper provides a clear answer to this question and demonstrates through methodological and empirical analyses that self-attention can be used as a stand-along primitive and also provides specific analyses, including different ablative analyses, showing under what circumstances attention can underperform or outperform convolution and under what circumstances an optimal combination between the two can be used for delivering the best performance. Empirical analyses are obtained using ImageNet for classification and COCO for object detection. The paper reads well and provides a good balance in methodological and empirical analyses and discussions. The findings and conclusions are interesting but not surprising. They are useful for computer vision practitioners and researchers. In this sense, the work is of both academic and societal impacts. Overall, I like this work and I am confident that the community shall benefit from these findings and conclusions from this work. I have a few minor questions on this work. - Presumably, convolution-stem + attention should deliver the best performance. Why on Table 1 for ResNet-50 is full attention better than convolution-stem + attention? The similar observations are obtained in Figure 3 for certain scenarios? - The conclusion that enlarging the spatial extent k in attention improves performance but plateaus off at 11x11 is based on the observations on Table 4. I wonder whether this conclusion is premature – what if you continue enlarging the spatial extent? Would the performance drop after 11x11? Or plateaus? Of increase again? - In general, the conclusion that convolution is good at capturing low level features while attention is good at higher level is probably valid for all the “natural” images like those in ImageNet and COCO. What if you have binary/illusory/sketch images where you may need attention in the first place? The presentation of the work is acceptable. But there are grammatical errors and typos. Also somehow most of the references were missing in the paper. The above was my initial review. I read authors' response and am happy with their answers. I stay with my original review.

Reviewer 3



The paper addresses the problem of replacing convolutions with self-attention layers in vision models. This is done by devising a new stand-alone self-attention layer, which borrows ideas from both convolution and self-attention. Like convolutions, it works on a neighborhood of the image, but replaces dot operations with self-attention operations. Unlike convolutions, this layers features a significant reduction in the number of parameters and computational complexity, plus the parameter count is independent on the size of the spatial extent. As in sequence modelling, they employ relative position embeddings, on both rows and columns of the neighborhood. Experiments are carried out on (almost) state of the art classification and detection architectures, where the authors replace convolutional blocks with their self-attention layer, and use average pooling with stride to do the spatial down sampling. As they have experimental findings that self-attention is not effective in replacing the convolution stem, they also build a replacement for stem layers which is devised by injecting distance-based information in the computation of the values of the attention. The rest of the experimental evaluation ablates the architectures by considering the role of the conv stem, network width, modifying the set of layers which are replaced with self-attention, and varying the spatial extent of the self-attention. Overall, the paper introduces a novel and significant idea. Investigating the role of self-attention as a stand-alone layer is a fundamental research question given the recent advances in both vision and sequence processing. The paper is also well written and clear. On the other side, a deeper investigation of what self-attention can learn with respect to convolution would have been useful, but is left for future works. The provided experiments underline that the new layer can significantly reduce the number of parameters and computational complexity, therefore the result is definitely interesting by itself and opens possibilities for future research.

[Author Response · NeurIPS 2019]

**Rebuttal for ID 41**. We would like to thank the reviewers for their time and thoughtful comments.

**[R1]** *"The goal of the replacing convolutions with local self-attention is a bit in contradiction..."* Self-attention has several advantages over convolutions, even when restricted locally.

- In contrast with convolutions where each position shares the same kernel weights, multi-head self-attention generates local kernels that can have different weights per position due to the computation depending on content-content interactions.

- Additionally, local self-attention is more parameter efficient than convolutions: using a $7 \times 7$ local self-attention layer outperforms using a $3 \times 3$ convolution while having $3\times$ fewer parameters. Furthermore, the $7 \times 7$ local self-attention layer has $2.4\times$ fewer FLOPs than a $3 \times 3$ convolutional layer.

**[R1]** *"it seems that the parameters of the convolutional layer have been replaced by the parameters ... $W_q$, $W_k$ and $W_v$."* We view the attention mechanism as a method for manufacturing convolutional kernels based on the content of a given location. In some sense, this is a relaxation of locally-connected layers by not requiring that the kernels be identical across spatial locations. The relaxation goes further by allowing the weights themselves to depend dynamically on the content of each image.

**[R1]** *"Why are positional features important for self-attention"* Without positional information, attention will not be sensitive to the ordering of the pixels because it will only use content-content interactions. Convolutions implicitly carry a relative positional encoding by having weights that depend on relative distance.

**[R1]** *"One could consider that using only the positional interaction is a degenerated form of convolution"* We agree that the importance of the content-relative interaction is surprising and concur in its similarity to traditional convolution, but expect that in future investigations on more challenging tasks than classification the relative importance of content-content interactions will increase.

**[R1]** *"throrough comparison of CNN and the proposed self-attention from a computational point of view [...] and the expected behaviour and properties.* For kernel size $k$, channels $d$, convolution cost scales as $k^2 d^2$ FLOPs per position with $k^2 d^2$ parameters, while self-attention cost scales as $3d^2 + k^2 d + kd$ FLOPs per position with $d^2$ parameters.

**[R3]** *"Why on Table 1 for ResNet-50 is full attention better than convolution-stem + attention?"* In the cases where full attention outperforms convolutional-stem with attention, the difference is small ($\leq 0.2\%$) and can likely be explained by variance in training runs. In the final version, we will add error bars to capture the variance.

**[R3]** *"...enlarging the spatial extent $k$ in attention improves performance but plateaus off at $11 \times 11$..."* We will experiment with larger $k$ in the final version. We suspect that the effect of changing $k$ is task dependent.

**[R3]** *"What if you have binary/illusory/sketch images where you may need attention in the first place?"* While this work is focused on demonstrating the attention can be used as a fundamental primitive for building vision models, studying the performance on different input domains is an exciting future direction, as is understanding the relative merits of convolution and attention beyond standard classification and detection tasks. Other study directions include benchmarking performance of convolutional vs. attentional models on transfer and self-supervised settings.

**[R3]** *"grammatical errors and typos. Also somehow most of the references were missing in the paper."* We apologize for accidentally clipping 4 pages of references section and any grammatical errors. We have addressed all of these issues and will restore the references in the revised manuscript.

**[R4]** *"downsampling is carried out with average pooling with stride 2... instead of increasing the stride of the self-attention layer"* We tried this downsampling approach in early experimentation and found it slightly underperforms compared to average pooling. However, this experiment was conducted on a preliminary architecture, so we plan on running experiments to benchmark this conceptually simpler approach on our final architecture.

**[R4]** *"not clear what self-attention can learn with respect to convolution, and what would happen with deeper models"* We agree that a more rigorous study of the modeling and optimization capabilities of attention and convolution would be illuminating. We leave this to future work. However, one clear difference is attention can generate a different kernel per position based on content, while convolution uses the same kernel for every position.

[Meta-Review · NeurIPS 2019]

The paper proposes a new model for the deep learning pipeline in vision, by replacing convolutions with self-attention layers in vision models. A new stand-alone architecture is proposed with good experiments. Initially two reviewers already proposed an acceptance while one reviewer asked for some improvement and explaination. The reviewers exchanged several comments and all agree on the fact that the answers in the rebuttal were convincing. Therefore, also for the meta-reviewer the final rate is “ accept”.